# Behavior and Task Classification Using Wearable Sensor Data: A Study across Different Ages [note 1]

**DOI:** 10.3390/s23063225

**Published:** 2023-03-17

**Authors:** Francesca Gasparini, Alessandra Grossi, Marta Giltri, Katsuhiro Nishinari, Stefania Bandini

**Affiliations:** 1Department of Informatics, Systems and Communication, University of Milano-Bicocca, 20126 Milan, Italy; 2RCAST—Research Center for Advanced Science & Technology, The University of Tokyo, Tokyo 153-8904, Japan

**Keywords:** wearable sensors, physiological signals, PPG, EMG, GSR, signal processing, classification

## Abstract

In this paper, we face the problem of task classification starting from physiological signals acquired using wearable sensors with experiments in a controlled environment, designed to consider two different age populations: young adults and older adults. Two different scenarios are considered. In the first one, subjects are involved in different cognitive load tasks, while in the second one, space varying conditions are considered, and subjects interact with the environment, changing the walking conditions and avoiding collision with obstacles. Here, we demonstrate that it is possible not only to define classifiers that rely on physiological signals to predict tasks that imply different cognitive loads, but it is also possible to classify both the population group age and the performed task. The whole workflow of data collection and analysis, starting from the experimental protocol, data acquisition, signal denoising, normalization with respect to subject variability, feature extraction and classification is described here. The dataset collected with the experiments together with the codes to extract the features of the physiological signals are made available for the research community.

## 1. Introduction

Urban and domestic environments are becoming increasingly related to technology. Being able to automatically interpret human behaviors and people’s feelings could provide significant insights for the definition of systems that can automatically react depending on the interacting person.

As a concrete example, let us consider a future scenario where self-driving vehicles are increasingly adopted in an urban environment. In the case of traditional vehicles, the establishment of eye contact with the driver provides a perception of safety to the pedestrian [1]. Considering self-driving vehicles, it is mandatory that they establish communication with the pedestrians, giving them effective feedback [2], and adapting their behavior (for instance, emitting an alert signal or reducing speed) with respect to the feelings and safety perception of every type of individual [3]. This is particularly crucial when dealing with the most vulnerable ones, such as older adults and people with disabilities [4].

Another example in an urban environment where technologies capable of receiving feedback from citizens could be involved is the realization of traffic lights able to adapt their waiting time according to the presence or absence of people with impaired mobility.

To realize such systems, an essential aspect consists in profiling and recognizing the categories of individuals with which these systems could interact. Within this context, subjects’ age is a relevant factor that should be considered. It has been shown that subjects of different ages react differently to stimuli both from an emotional [5] and behavioral point of view [6,7]. For instance, older adults appear less reactive than young adults in response to audio and visual stimuli [8], as well as appearing slower in carrying out cognitive tasks such as mouse pointing [9] or in driving ability [10].

Nowadays, physiological responses have become useful to better understand how different types of people react to different situations. They can be considered reliable indicators of uncontrolled and fair human reactions to external stimuli and are now widely used to detect and recognize affective states [11] and human behavior in different environments [12], and to detect different human emotions [13,14].

Nowadays, the significant improvement in wearable sensor technology and the progressive reduction in their cost allow their adoption in many new experimental scenarios, making them comfortable also for older adults or impaired subjects. Their utilization, as opposed to the deployment of more complex but more invasive devices, allows the recording of physiological signals during, for example, daily life activities [15], without hindering the subjects’ capability of performing tasks, giving them much more freedom in terms of movement. However, physiological data coming from wearable sensors need to be carefully tested through observations, interviews and rigorous experiments, both in real-life scenarios and in formally designed experimental sets, namely, under laboratory conditions [16]. Another important factor that should be taken into consideration is that the physiological response of every person is diverse because of how we are naturally different from each other: we are different in terms of biology, memories, cultural influences, experiences and personality and, depending on the environment, even the same subject could react to the same stimulus in different ways. These considerations render it necessary to investigate how physiological responses are related to subjects, stimuli and environments, and how we can classify them in order to perform proper and universally effective recognition tasks. Given the importance of understanding interactive human behavior, both with other human beings and with the environment, the first contribution of this work is the data collection of physiological signals using wearable sensors through an experiment that involves subjects of different ages performing various tasks. In particular, data are collected considering time varying stimuli, changing the cognitive stress of the subjects, as well as space-varying environments, changing the walking conditions of subjects and introducing collision avoidance tasks. During the experiment, the participants wore sensors that measured their heartbeats through photopletysmography (PPG), galvanic skin response (GSR) and muscle activity using noninvasive electromyography (EMG). The PPG and GSR sensors are placed on the fingers of the dominant hand, while the EMG sensors are placed on one leg. The sensors used are noninvasive and completely painless.

Relying on the data collected, the main research questions of this paper are:Q1: Can physiological data, acquired with wearable sensors, be useful to define binary classification models that allow discriminating between young adults and older adults while they perform various tasks?Q2: Can a binary classification model, trained on two tasks of different cognitive load, be sufficiently generalizable to classify a new set of different binary tasks?Q3: Can physiological data be useful to discriminate contemporaneously among different tasks and ages?Q4: Can physiological data, potentially affected by movement noise, reveal the different behavior of subjects with respect to their age and walking activities?

Research questions Q1, Q3 and Q4 will give hints about the feasibility of defining interacting systems that react differently with respect to the peculiar reaction of different individuals. Q2 is crucial when considering that real-life activities are variegate and it is not reasonable to imagine that for each of them a classification model should be trained. The requirement of generalizable models is thus mandatory. This paper is structured as follows: In Section 2, an overview of the state of the art regarding the use of physiological data in classification and behaviour analysis is reported. In Section 3, our experimental protocol and data collection are described. In Section 4, data pre-processing required to denoise, normalize and segment raw data is presented, together with the classification models and validation strategies adopted. In Section 5, the results of the analyses performed on data collected are reported in detail and discussed in Section 6. Finally, conclusions are drawn and future work is presented.

## 2. State of the Art

Physiological signals are regulated by the parasympathetic and sympathetic systems, and thus are not directly controlled by the subjects and can be considered honest indicators of emotions and mood [17,18]. For this reason, physiological signals have been widely used in the recognition of human emotion [19], in understanding subjects’ behavior during daily life activities such as safe driving [20], in health care applications, for instance to monitor mental health [21] and in the social security field [22]. In the literature, a variety of different stimuli have been adopted to induce changes in human emotions and their affective state, ranging from audio visual ones, such as audios with different music genres [23], to emotional evocative music videos [24,25] and cognitive stimuli such as math calculations [26,27]. Physiological signals have been successfully used to analyze the change in arousal and to detect stress. In particular, heart rate and skin conductance appear very promising in arousal detection [18,26]. Arousal is an uncontrolled human reaction, linked to attention and cognitive readiness, activated by stimuli that require a high psycho-physical commitment, and therefore activated in particular during cognitive tasks and stressful conditions. In [28], it is reported that arousal of pedestrians during crossing is affected by road characteristics, such as the number of lanes or the density of vehicles, as well as by the age of the pedestrian himself/herself. In the field of urban mobility, the analysis of physiological data acquired by subjects moving in different settings has allowed inference of environmental conditions perceived as stressful by people with visual impairments (VIP) [29,30] or to automatically detect arousal induced in a subject by different environment characteristics [31]. Physiological signals are also used to identify different levels of cognitive load while performing mental tasks. It is proved that high level of mental stress due to excessive workload can negatively influence individuals’ health, bringing, in extreme cases, physical and mental illness [32], and thus a continuous monitoring of workers stress levels can be useful to allow prompt actions. A survey of the main stressors adopted during cognitive tasks is reported in [33]. Arithmetic mental calculus and the Stroop test are very powerful methods to elicit mental stress, and have been applied in numerous experiments [34,35]. In many of these studies, physiological signals allow one to discriminate well between different levels of cognitive load, proving their goodness both in statistical [36] and classification analysis [37].

Cognitive load has been widely analyzed in the study of an individual’s workload during a driving task. In this context, physiological signals have been considered to monitor the level of driver physiological arousal in response to cognitive load during car driving [27,38], to evaluate the level of mental workloads in subway driver operators [39] or to assess the mental workload in an aircraft pilot during flights [40,41].

Concerning human activity recognition (HAR), the use of physiological signals combined with inertial and position data have shown positive results in the recognition of the person’s activities, even during movement tasks such walking or running [42]. In this context, many studies focus their attention on the analysis of heartbeat collected by PPG. The use of this type of signal, however, is limited by the sensitivity of the PPG sensors to motion artifacts (MAs) that could lead to an erroneous heart rate estimation [43]. In the literature, several denoising strategies based on filters have been proposed to face this problem [43,44,45]. In [46], instead, it is reported how the presence of these artifacts can be useful to discriminate different activities and, thus, features extracted from motion artifacts can be used, in addition to other PPG features, to predict the types of activities performed by users. In general, physiological signals seem positively correlated with the intensity level of the activity [47], allowing one to discriminate among several human activities such as sitting, walking, jumping and running, but seem less effective to recognize tasks such as jumping and jogging, cycling and walking or sitting and standing [46,48].

Finally, in most of the studies reported so far, the populations considered are uniform with respect to age. However, physiological signals, as well as how a person reacts to stimuli or conditions, change with the age. In [49], for instance, it is reported how young adults and older adults present differences in emotional reactivity when positive or negative emotions are elicited by movies. The authors of [50,51] reported that the average heart rate of a subject as well as his/her heart rate variability tends to decrease with age. In [52], it is shown how the gait pattern of a subject during different cognitive load tasks tends to become less regular with the increase in the subject’s age. A further difference between young adults and older adults concerns the way in which the two populations react to potentially dangerous situations. In [28], it is reported how senior pedestrians tend to underestimate the crossing time, having a higher amount of stress when crossing as compared to young people. Similarly, in [6], it is shown how older adults tend to keep a more careful behavior than young people when they have to avoid an obstacle, passing only in safety conditions. Only a few analyses [53] have been performed in recognizing the subject’s age using physiological signals, making this issue an area of research that is still open.

## 3. Material and Methods

### 3.1. Experimental Protocol

The CLAWDAS (Cognitive Load and Affective Walkability in Different Age Subjects) dataset here presented has been collected in a controlled laboratory environment at the Research Center for Advanced Science and Technology (RCAST) at the University of Tokyo. In the experiment, two different groups of subjects were involved: a population of young adults, composed of 16 Japanese master and PhD students, (average age = 24.7 years, standard deviation = 3.3, 4 women), and a population of Japanese older adults (retired), 20 subjects, (average age of 65.15, standard deviation = 2.7, 10 women). Mental or heart diseases could influence the physiological responses, thus the healthy conditions were considered as inclusion criteria for the selection of the participants.

All the participants performed the same tasks defined by the experimental protocol, lasting about one hour, and consisting of two sessions: a first part in which subjects sat at a desk performing several cognitive load or relaxing tasks, and a second part in which, instead, several walking conditions were proposed. The experiments in this paper have been reviewed and approved by the Research Ethics Committee at The University of Tokyo, Japan (No. 19-283 and 19-376).

The two sessions of the experiment are described in more details below:

Cognitive Load session: lasting about 30 min and composed of the following steps:Subject’s profiling carried out filling the STAI questionnaires [54], 3 min.Reading and comprehension (R, C) tasks. Two different texts are proposed: a Fairy-tale (“The Wolf and the Seven Little Kids”) and a philosophy text (“Kant’s Critique of Pure Reason”). The subjects have 2 min to read each text on a piece of paper and 1 min to answer self-assessment and reading comprehension questions, 6 min.Audio listening and math calculation (AL, MC) tasks composed of six repetitions of a two-step sequence consisting of:
Audio listening, performed using headphones, in which the relaxation is induced by natural and real-life sounds (Figure 1 right), 2 min.Mental arithmetic calculations such as sums, subtractions and multiplications (Figure 1 left), 30 s.
The audio tracks and the calculations proposed are the same for each subject but change according to the iteration, 15 min.


Between each couple of tasks, a period of resting time (baseline acquisition B${}_{C}$) of about 1 min is acquired.

The audios and math calculations were chosen following the results obtained in [12] in order to induce equal levels of relaxation and equal levels of cognitive load. There was no intention of inducing several levels of cognitive load intensity. The math calculations chosen were: (i) 27 × 5 − 20, (ii) 139 − 21 + 17, (iii) 24 × 4 + 101, (iv) 39 × 4 + 2, (v) 27 × 9 − 11, and (vi) 54 + 19 − 31. The audios were selected as inducing a high level of relaxation. They were chosen among several audios also adopted in the previous study and were evaluated by three subjects. The audios are: (i) chirping of birds, (ii) lapping water, (iii) rain, (iv) sea waves, (v) waterfall and (vi) wind.

Walking session: Three within subject walking conditions are considered:Collision avoidance: two subjects, at the same time, walk with their own pace along a *U* path (Figure 2 top left). At about half of the path, they reach the collision avoidance zone where they have to avoid the collisions with both the obstacles (Obs) and the other subject (Figure 2 top and bottom right). Then, they complete the *U* path, with their natural pace (WO), and go back in the opposite direction repeating the same actions. The obstacle is a moving pendulum composed of a swinging mattress activated manually by one of the experimenters.Forced speed walk: the participants walk with a forced speed based on the metronome ticking. Three speeds are considered: 70 bpm (F1), 85 bpm (F2) and 100 bpm (F3). The idea of adopting different walking paces was inspired by previous work on pedestrian flow analysis [55,56].Free walk (FW): the participants walk freely, without constraints, following their own pace.

Between each couple of tasks, a period of resting time (baseline acquisition, B${}_{W}$) of about 1 min is considered. The whole procedure is repeated three times.

### 3.2. Physiological Signals Acquired

During the whole experiment, physiological signals are collected using wearable sensors from the Irish company Shimmer (www.shimmersensing.com). These low-cost wearable sensors were already utilized in different experiments concerning physiological signal analysis and affective state recognition with encouraging results [57]. In this experiment, the Simmer3 GSR+ unit and the Shimmer3 EMG unit are adopted, and Figure 3 shows how they are worn by the subjects during the experimentation. The physiological signals acquired are: galvanic skin response (GSR), also known as skin conductance (SC), which is connected to sweating and perspiration on the skin and is a reliable stress indicator, photoplethysmography (PPG), which measures the blood volume registered just under the skin and can be used to obtain the heart rate of the subject, surface electromyography (EMG), which measures the muscle activity of the medial gastrocnemius muscle and the anterior tibial muscle. Inertial data are also collected through accelerometer and gyroscope sensors. PPG and GSR signals are collected using a sampling frequency of 128 Hz while the EMG signals were acquired using a sampling frequency of 512 Hz. The adopted sensors are shown in Figure 3. EMG signals are acquired only from 8 male subjects for young adults, while from 10 subjects (3 females and 7 males) for older adults. A first visual inspection of the physiological signals acquired during the experiment was performed by the experimenters in real time using Shimmer’s proprietary software Consensys. This operation allowed us to detect in time those signals which would have been unusable due to high noise or poor adherence of the sensor to the subject’s skin, and immediately repeat the task if necessary.

In Table 1, the number of instances collected for each sensor and task in both cognitive and walking sessions are reported. Note that in the case of young adults, the free walk task (FW) is not acquired, and the collision avoidance task (WO + Obs) is not segmented separately into (WO and Obs).

## 4. Data Analysis

In order to be adopted in a classification model, raw physiological signals should be: (i) processed to reduce noise, (ii) normalized to disentangle subjects dependencies and (iii) segmented into different tasks. The data processing algorithms adopted in this work are described in detail in the next section (Section 4.1). Once raw data have been properly processed, features that describe signal patterns can be extracted and adopted in classification models. Machine learning and cross validation strategies considered in this work are described and detailed in Section 4.2.

### 4.1. Data Processing

Data processing algorithms applied to raw data are detailed in the following sections:Signal denoising: each signal can be affected by different types of noise and artifacts related to both the characteristics of the environment (for instance, temperature or electromagnetic interference) or the experimental conditions (for instance, uncontrolled movements of the subjects, or low-quality contact of the sensor with the subjects’ skin). Thus, in Section 4.1.1 the proper denoising for each physiological signal is reported.Subject normalization: physiological signals not only depend on the induced stimulus, but also on subject’s characteristics. A proper normalization process is performed to reduce both inter- and intra-subject heterogeneity, as described in Section 4.1.2.Signal segmentation: data are acquired continuously; thus, a proper segmentation is applied, adopting the markers introduced during the acquisition through Consensys Pro, the proprietary software of the Shimmer devices.PPG frequency normalization: in order to take into account frequency differences in subjects’ heartbeat, a frequency normalization strategy is applied to segmented data.Data augmentation: to balance the cardinality of the instances per classes, a proper data augmentation step is applied as described in Section 4.1.5.

All the analysis and pre-processing operations are performed using MATLAB 2020a.

#### 4.1.1. Signal Denoising

The raw signals collected during the experiment appear corrupted by noise and artifacts that could make them unusable or difficult to analyze. These artifacts could be due to the characteristics of the environment where the signals are acquired (ambient or artificial light, electromagnetic interference or temperature), to the errors in experimental setup (poor cohesion between the sensors and skin or artifact owing to subject’s breathing or movement) or to the influence of other physiological processes in the body (the activity of neighboring muscles in the case of the EMG analysis) [58].

In order to remove these artifacts, the PPG raw signals are pre-processed using a multiresolution wavelet denoising strategy described in [6], and in [59]. In this method, each signal is divided in frequency sub-bands using stationary wavelet transform [60] with mother wavelet Fejer-Korovkin and with four levels of decomposition. A soft thresholding is then applied to the detail coefficients of each sub-band using the universal threshold calculated by the formula $t_{k}=\sqrt{2log(N_{j})}$, where $N_{j}$ is the length of the *j*th wavelet coefficient [61]. Finally, the signal is reconstructed using the inverse of the stationary wavelet transform previously described. The algorithm used to implement the SWT in MATLAB is the “algorithm a-trous” introduce by [62]. In order to use this method in our study, a preliminary operation of replicate padding is applied to each of the PPG signals to obtain signals with length divisible by $2^{4}$, where four is the number of levels used in the decomposition [60].

A similar denoising strategy based on multiresolution wavelet decomposition is adopted in pre-processing GSR signals. Following the positive results achieved in [63], a stationary wavelet transform with Coiflet (Coiflet3) as a mother wavelet is used to divide each of the GSR signal in seven levels of decomposition. Then, a soft thresholding operation is applied to each of the detail coefficients using, in this case, a fixed threshold determined trying to yield the minimum of the maximum mean square error over a given set of functions (minimax thresholding method). Even in this case, the use of “algorithm a-trous” to implement the stationary wavelet transform made it necessary to first apply the replicate padding algorithm to each of the GSR signal in order to make the length of each signal divisible by $2^{level}$, where “*level*” in this case is equal to seven.

Finally, in the literature it is reported how the use of a multiresolution wavelet denoising strategy appears promising in removing noise also from EMG signals [64,65]. In particular, in our study, a multiresolution wavelet denoising method using maximal overlap discrete transform (MODWT) with a mother wavelet Daubechies-4 and five levels of decomposition is adopted to pre-process both the EMG channels collected in the dataset (tibial and gastrocnemium). Similarly to PPG, the universal threshold is considered during the soft threshold operations.

#### 4.1.2. Amplitude Normalization

For each analyzed signal, the denoising operations are followed by a normalization task performed to reduce both the inter- and intra-subjects’ heterogeneity. Concerning the PPG, the amplitude of the whole subject’s signal for each session is standardized by the application of the Z-score operation. Therefore, the mean and standard deviation of the whole subject session PPG signal are computed and used according to the formula Z=(x−μ)/σ. Likewise, following [66], the GSR signal of each subject is also normalized in amplitude using z-scoring. Even in this case, the mean and the standard deviation of each subject’s session signal are computed and used for re-scaling the whole signal of that session. In case of EMG signals, instead, the amplitude normalization is carried out dividing each channel of the denoised signal by the maximum peak activation value obtained from the signal itself. This strategy, described in [67], has been selected, after an empirical analysis, as it was the most effective method to reduce the inter-subjects’ variability. Even in this last case, the amplitude normalization is applied considering the signals related to each session separately.

#### 4.1.3. Signal Segmentation in Task

Both the denoising and amplitude normalization are applied to the whole signals collected from each subject during the two different sessions (cognitive or walking). Therefore, a segmentation operation is then performed to divide each signal into the different tasks as defined by the experimental protocol. In particular, for each participant, the markers manually recorded during the data acquisition phase are used to determine the beginning and end of each trail performed by the subject and, thus, to define the edge of the segments in which the session signal has to be divided. In Table 1, the number of signals for each task resulting from this segmentation are summarized. In particular, the first two rows of the table refer to PPG and GSR signals while the last two refer to gastrocnemium and tibial EMG signals.

#### 4.1.4. PPG Frequency Normalization

The normalization applied so far to PPG signals allows us to uniform the signals concerning amplitude but does not take into account the differences in the subjects’ heartbeat that are more related to frequency. In [68], it is reported how the heart rate frequency of a resting adult can vary in a range between 60 and 100 beats per minute, depending on different factors both personal (such as age, sex, ethnicity, sports ability, diet, illnesses, prescribed medications, etc.) and environmental (humidity, temperature, etc.). To this end, only for PPG signals, the segmentation phase is followed by a frequency normalization phase, performed in order to reduce the heterogeneity in signals related to heartbeat differences. This normalization is carried out using a new subjective resampling frequency calculated started from each subject’s baseline heartbeat. Thereby, each PPG signal, originally defined in a discrete time domain (DTD), is mapped into a new subject-normalized discrete domain (SND) where all the subjects are normalized with respect to their heart beat in a resting state condition. More details about this normalization can be found in [69].

#### 4.1.5. Data Augmentation

In order to increase the cardinality of the data and create more balanced classes for the next classification step, some segmented signals are divided into non-overlapping segments. In particular, the signals collected during the reading task (R) are divided into segments of 40 s using non-overlapping windows of fixed length, while the reading comprehension signals (C) are segmented into two parts of equal length. Thereby, the number of signals for reading task increases from 32 to 96 for young adults, and from 32 to 64 for older adults. Similarly, the cardinality of the signals related to the comprehension task increases from 40 to 120 for young adults and from 40 to 80 for older adults.

A similar strategy is also adopted for the signals related to the free walking (FW) task in order to create more balanced classes during the classification analysis. In particular, all the instances collected during the pure free walking task are divided into two segments of equal length, increasing the cardinality of each analyzed physiological signal from 57 instances to 114.

### 4.2. Classification Strategies and Performance Evaluation

In this section, the classification settings adopted in the next data analysis are detailed. In particular, in Section 4.2.1, features extracted for each physiological signal analyzed here are reported. In Section 4.2.2, the classification models and cross validation strategy adopted are discussed, while in Section 4.2.3, the metrics adopted to evaluate the performance of the classification models are introduced and justified.

#### 4.2.1. Feature Extraction

For each analyzed signal, several features are extracted in order to highlight significant characteristics that could describe the subject’s physiological behavior in the different tasks. Considering the PPG signals, seven time-domain features are calculated:Four statistical features (minimum, maximum, mean and variance of the signal)Three peak related features:
–Peak rate, representing the mean number of peaks of each 128 subject-Normalized sample;–Inter-beat interval (IBI), representing the mean distance between two peaks in a row;–Root mean square of successive distance (RMSSD) representing the quadratic mean of the distance between two peaks [70].


In the analysis of GSR signals, both the phasic and tonic component time domain features are considered. Following [71], the GSR signal could be decomposed into three parts: the phasic componentthat represents rapid changes in skin conduction (skin conductance responses) due to external stimuli or spontaneous responses, the tonic component related to the slow change in the signal and representative of the general arousal level and the additive white Gaussian noise term incorporating errors and artifacts. In our analysis, the GSR signals are first decomposed into these three components applying the Cvx algorithm described in [71], made available at the MathWorks File Exchange repository by [72]. Then, several features are computed for the phasic and tonic components, as listed below:
Phasic Component: eight statistical and peak-related features:
–Maximum, mean and variance of the phasic component of the signal–Peak rate, representing the mean number of peaks per second–Peak area and peak area per second, representing, respectively, the mean area under the peaks and the mean area under the peaks evaluated per second.–Peak height representing the mean height of the peak detected on the phasic component.–Rise time (or also onset-to-peak time) defined as the mean number of samples from the onset of the skin conductance response to the top of the peak [73,74]
Tonic component: the Regression coefficient is considered as representative of the signal slope.


The MATLAB codes to calculate all these features are available together with the dataset.

For EMG signals, two features are considered:The mean power of the signal, calculated by the Root Mean Square [75]
(1)$$\mathrm{RMS}=\sqrt{\frac{1}{N}\sum_{n=1}^{N}x_{n}^{2}}$$
where $x_{n}$ is the amplitude of the *n*-th sample of the EMG signal, and *N* is the total number of samples.The walking frequency, known as the Stride Frequency, evaluated in terms of the number of steps per second. This feature is calculated as deeply described in [76] and here summarized by the following steps: (i) the root mean square upper envelope is calculated with 200 sample windows, (ii) the mean value is subtracted to rescale the signal, (iii) the envelope signal frequencies in the band [0.2, 1.4] Hz are considered, as representative of human pace frequencies and (iv) the max peak of the periodogram of the envelope is extracted.

Many of the data analyses reported here consist in classification tasks aimed at recognizing the age of the subjects or the levels of arousal induced by the different tasks performed. In these analyses, only the features related to PPG and GSR signals are considered. In particular, three different feature sets are taken into account during the classification tasks: features extracted considering only PPG signals, features extracted considering only GSR signals and the joining of PPG and GSR features. It is important to underline that all the features are standardized using the z-score before using them as an input to the different classifiers. For each experiment, this standardization is applied to the whole set of analyzed instances before splitting it into training and test sets.

#### 4.2.2. Classification Models

Two well-known classifiers frequently used in the literature to classify physiological signals are trained using the standardized features: a classification and regression tree (Cart) [77] and a support vector machine (SVM) [78]. Concerning the Cart classifier, the Gini’s diversity index is used as the criterion of splitting while the max number of the decision split is set equal to 100. Three different kernels in the case of the SVM are tested: linear (SVM-Linear), gaussian (SVM-Gauss) and polynomial cubic (SVM-Cubic) kernel. In our analysis, the kernel scale of the Gaussian kernel SVM is set equal to 3.3 in order to consider a medium Gaussian SVM for all the analyses performed.

To evaluate the performance of the trained classifiers, a leave one subject out (LOSO) cross-validation strategy is employed. In this strategy, the analyzed data are divided into subsets according to the subject from which the data were acquired. At each iteration, all the instances of one subject are used to test the model, while the signals of the remaining subjects are used as a training set. This method allows us to generate more robust performances when considering new unseen data, as it prevents the classifier from being trained and tested simultaneously with instances acquired on the same subject [79]. The confusion matrices resulting from each iteration are then joined and an overall final confusion matrix is thus generated.

#### 4.2.3. Evaluation Metrics

Several metrics are computed in order to evaluate and compare the performance of the different classifiers. In particular, in addition to the traditional and well-known accuracy, precision and recall, a single class F1-score [80] is evaluated as the harmonic mean of single-class precision and recall. The formula used to compute this metric is:(2)$${F1\mathrm{-score}}_{c}=2*\frac{precision_{c}*recall_{c}}{precision_{c}+recall_{c}}$$
where *c* is the class considered while $precision_{c}$ and $recall_{c}$ are the corresponding metrics evaluated considering as positive the elements of the *c* class. The F1-score metric gives a general hint in evaluating the classifier performance in recognizing a single class. As a harmonic mean of precision and recall, it provides information about the ability of the classifier in balancing precision and recall for the considered class, avoiding high recall at the expense of low precision and vice versa.

Starting from the single class F1-score, we calculate the weighted F1-score metric ($W-F_{1}$) using the following formula:(3)$$W-F_{1}=\sum_{j=1}^{m}\frac{N_{c}}{N_{tot}}*F_{1j}$$
where *m* is the number of classes considered, $N_{c}$ is the number of elements in class *c*, $N_{tot}$ is the total number of elements analyzed and $F_{1j}$ is the F1-score for the *j*-th class [81]. The $W-F_{1}$ is evaluated as the weighted mean of per-class F1-scores. It is a robust index of the general performance of the classifier as it weighs the classifier performance for each class, according to its number of elements in the dataset. To better quantify the performance of the proposed classifiers, in the next analysis we report as baseline the performance of random classifiers.

## 5. Results

In this section, we report the results of the analyses performed on data collected in the two different experimental sessions, trying to answer the four research questions:Q1: Can physiological responses discriminate between young adults and older adults while they perform different tasks?Q2: How much is a binary classification model driven on physiological data able to generalize on data collected while performing different tasks?Q3: Is it possible to define a multi-class classifier, driven on physiological data, that recognizes both the tasks and the age of the subjects simultaneously?Q4: Can physiological data, potentially affected by movement noise, reveal the different behavior of subjects with respect to their age and walking activities?

In particular, for what concerns the cognitive load session, the following classification tasks are considered:Population age classification using PPG and GSR signals. Four binary classifiers are proposed to discriminate between young adults and older adults, performing four different activities (Section 5.1.1). This analysis tries to answer research question Q1. A multi-class model is also developed to distinguish among six classes, obtained considering two population ages and three activities. This issue tries to solve research question Q3.Cognitive task classification. A classification model is trained on data acquired during math calculations and the relaxing audio listening (Section 5.1.2).Binary classification of new cognitive tasks using the previously trained model (Section 5.1.3). These last two items try to answer research question Q2.

Considering data collected during the walking session, the following analyses are performed to answer research question Q4:Different walking behavior recognition through muscle activities and PPG analysis (Section 5.2.1 and Section 5.2.2).Classification of different walking tasks, with similar walking pace (Section 5.2.3).

### 5.1. Cognitive Load Session

#### 5.1.1. Population Age Classification Using PPG and GSR Signals

The first analysis aims at evaluating if it is possible to distinguish between young and older adults, considering their physiological signals acquired during tasks that imply different cognitive load.

To this end, four binary classification tasks are performed, one for each activity (reading R, comprehension C, math calculations MC and audio listening AL). In all the classifications, two classes are taken into account: a class made up with the signals collected from young adults and a class related to the signals collected from older adults. In order to create balanced groups, the data augmentation strategy described in Section 4.1.5 is applied to the reading and comprehension signals. The classifiers trained for each analysis are evaluated in terms of weighted F1-score using the LOSO validation strategy, achieving the results summarized in Table 2.

In general, the best performances are obtained considering both PPG and GSR signals, allowing to reach values of an *W-F*${}_{1}$ score usually greater than 70%. The math calculation, MC, is the task in which the two populations are best distinguished, with a value of a *W-F*${}_{1}$ score of 78% obtained using SVM-Linear and PPG and GSR features together. The task where the classifiers generate the lowest performance, on the other hand, is the audio listening task, AL, where the *W-F*${}_{1}$ score of 67% is achieved using the SVM-Linear, trained only with GSR signals. In each task analyzed, the overall best performance is obtained using the SVM with linear kernel while, in general, the lowest ones are recorded using the Cart classifier.

As a further experiment, a multi-class classification analysis is conducted in order to discriminate not only the population group with respect to age but also the task performed. Therefore, six classes are considered: three activities (MC, AL, and R), for the two population groups. The comprehension task, C, is not considered due to its lower cardinality, even in the case of data augmentation. Only the SVM with linear kernel trained using the union of PPG and GSR features is selected for this analysis as it is the classifier that allows us to reach the best performance in recognizing the two populations in the previously described binary classification tasks.

The confusion matrix produced by this classifier is reported in Table 3. From the results, it emerges how the classes better recognized are the ones related to older adults subjects, where both the age and the tasks performed appear well discriminated by the classifier. On the other hand, the SVM with leaner kernel seems to struggle more in recognizing the signals collected from young adults with a percentage of recognition usually near to 50%. Finally, in case of misclassification, the algorithm tends to well classify the task performed, but to misunderstand the age of the population. For more details about this analysis, please refer to [82].

Due to the positive results achieved in discriminating young adults from older adults, we have decided to consider the two population groups separately in the next analyses.

#### 5.1.2. Cognitive Task Classification: Math Calculation vs. Audio Listening

The second analysis on data collected during the first session of the experimental protocol aims at defining classifiers able to discriminate between different cognitive load tasks. In particular, binary classification models are here investigated to recognize math calculation, MC, from audio listening tasks, AL. For each one of the two populations analyzed, the four classification models described in Section 4.2 are considered, varying the set of features used (PPG, GSR or joining PPG and GSR). The LOSO strategy is adopted for cross-validation, achieving the results reported in Table 4 in terms of accuracy, per-class F1-score and weighted F1-score.

The performances obtained for the two analyzed populations appear very similar when the same set of features (PPG, GSR or PPG and GSR) are considered. In fact, in both young adults and older adults, the best performances are reached using the features extracted from GSR with accuracy values around 92%. In particular, in young adults, two classifiers (SVM-Linear and Cart) are able to achieve the highest accuracy value of 93% using GSR. Similarly, in older adults, the use of GSR features allows to reach the highest accuracy of 92% using either SVM-Linear or SVM-Gauss. Although positive, the lowest performances are observed in the classifiers trained using only features extracted from PPG signals. In this case, the use of SVM with linear kernel allows us to reach accuracy values around 80% in both young adults and older adults. Moreover, analyzing the per-class F1-scores, it emerges that both the two classes are well recognized, with similar values in the performance metrics.

Finally, another interesting consideration regards the classifier that allows us to reach the best performances. In general, the use of SVM with a linear kernel allows us to achieve the highest accuracy value in almost all the conducted experiments. It is important to stress, however, that the same accuracy values are also reached by more than a classifier in the analysis using features computed from GSR. To this regard, for instance, in both populations, the Cart classifier generates the same performances of SVM-Linear when trained with the union of PPG and GSR features, as well as when trained with the GSR features extracted from the young adult signals. Likewise, in older adults, the two classifiers SVM with a linear and Gaussian kernel generate the same accuracy value when the features extracted from only GSR signals are considered. In general these results seem to confirm what is reported in the literature about the importance of the GSR in discriminating tasks having different levels of arousal [18,83].

#### 5.1.3. Arousal Classification of New Cognitive Tasks Using Pre-Trained Binary Classifiers

Starting from the results obtained in the previous section, this analysis aims to evaluate the performance of the classifiers trained on MC and AL tasks in recognizing new tasks that imply different cognitive loads. In [33], it is reported how a mental arithmetic task is a powerful and efficient method to induce stress in subjects, and is therefore used as a stress inducement stimuli in many stress-related analyses. Concerning the listening task, the audio tracks proposed to participants are natural sound effects chosen in order to induce a state of relaxation in people. These characteristics of the chosen audio tracks are confirmed by the results extracted from the self-assessment questionnaires collected during the experiment, where the majority of the audio tracks are considered by the subjects as inducing medium or high levels of relaxation. According to these considerations, the classifiers able to automatically recognize the two tasks could be also used to discriminate between high arousal (or equivalently high stress) and a low-arousal task (or equivalently low stress), represented, respectively, by math calculation and audio listening.

In the following analysis, we tried to apply these classifiers to the signals collected during different cognitive activities such as, for instance, reading and comprehension. The idea is to automatically evaluate, using the classifier, the level of arousal induced in the subjects performing these two tasks and, at the same time, to evaluate the possibility of applying a classifier trained on known data to new unseen ones, collected during different tasks. In particular, the following three tasks have been considered: the baseline B${}_{C}$, the reading task, R and the reading comprehension one C. In this part of the analysis, the signals related to R and C tasks are analyzed without data augmentation.

According to the results obtained in Section 5.1.2, only SVM-Linear is considered as it achieves the highest performance in recognizing the high-arousal task from the low-arousal one. In particular, in case of equal performance obtained from more than one classifier, only the SVM-Linear is considered in order to make the results as comparable as possible among the different analyses. Unlike the previous analysis described in Section 5.1.2, all the signals of math calculations and audio listening tasks are used to train the classifier without dividing them into training and test sets.

The classification model so trained is then employed to classify the unseen data collected in the B${}_{C}$, R and C tasks. The results generated are summarized in Table 5, where the percentage of instances predicted in each of the two classes (high arousal and low arousal) is shown according to the task performed, the population group and the physiological signal analyzed.

In all the analysis, most of the instances collected during the baseline are classified as inducing low arousal with percentages slightly higher in older adults (values usually greater than 90%) with respect to young adults (values usually around 83%). On the other hand, the instances related to text comprehension are mainly classified as high arousal with percentages usually between 70% and 85% in both the populations analyzed.

Regarding the reading task, the percentage of instances predicted as inducing low or high arousal appears more balanced, especially in the case of older adults, while in the case of young adults there is a predominance of instances predicted as inducing low arousal. This difference could be related to the higher difficulty of older adults in reading long text because of age-related factors (e.g., reduced visual abilities) that could lead to a higher cognitive load and, thus, to a greater emotional alteration. It is also worth noting that the young adults are students used to reading difficult texts.

### 5.2. Walking Session

The first analysis on data collected during the walking session relies only on EMG signals, which can reveal different walking behaviors through the analysis of muscle activities. Another important aspect investigated in this section regards the contributions in the PPG variations related to movement and those related to changes in arousal due to an increase in stress. Finally, a task classifier is proposed to evaluate if it is possible to recognize different tasks, despite the interference of movement with the acquired signals.

#### 5.2.1. Walking Behavior Studying EMG Data

In this section, the three different tasks of the walking session are analyzed using the EMG features described in Section 4.2 and a comparison between young adults and older adults behavior is performed.

From the analysis of the estimated frequency stride during the forced speed tasks, reported in Table 6, we observe that the older adults struggle more than young adults to respect the metronome forced speeds. In particular, this behavior is observed mainly in the two lower speeds, F1 and F2 where the frequency stride values detected on the older adult signals appear usually higher than the metronome ticking and more similar to the subject’s habitual cadence.

In the case of the fee walk task FW, the mean frequency stride detected is usually around 0.90 steps/s, appearing, in this way, in agreement with the metronome frequency indicated by the participants as the most preferred one, F3 and with the normal pace speed reported in the literature (between 0.90 and 1 steps/s as reported in [84]).

In the collision avoidance task, a stride frequency evaluation and a signal energy analysis is performed. Analyzing the obtained stride frequency values in the portions of the path far away from the collision zone, WO (see Figure 2), we obtained values in the range of [0.80–1] steps/s, similar to the values reported in the literature in the case of free pace and also detected during the free walk task of our experiment. However, compared with these latter values, we observe a greater variance. To better understand how the walking pace changes within the collision avoidance zone, an analysis based on signal energy is finally performed. In this analysis, we study how the muscle power detected in the EMG signals changes during the two phases that compose the task: the free walking phase before and after the collision avoidance zone WO and the crossing itself, Obs. This study highlights a different behavior between young and older adults during the crossing. In fact, when young adults are involved, an increase in the signal power is noticed in general in correspondence to the collision-avoiding events. Most of the time this growth seems due to strong muscle activation, probably caused by the effort of the subject to accelerate and safely pass the obstacle. On the other hand, from the analysis of the power of the EMG signals for the older adults, a decrease in signal power is observed during the collision avoidance events. These decreases are related to the observed evidence that the participants tend to decelerate or even stop in front of the obstacle, waiting for the pendulum to pass. More details about this analysis can be found in [6].

#### 5.2.2. Walking Behavior Studying Ppg Data

To compare the different walking tasks and to evaluate the effect of walking pace on heartbeat, a statistical similarity analysis is performed. To this aim, the non-parametric Kruskal Wallis test is applied [85]. This test is based on the null hypothesis whereby the two distributions provided as input are similar if their medians are equal. On the other hand, if the two distributions are different, the *p*-value returned by the test appear lower than a certain significance level (usually set to α = 0.05) and the null hypothesis is rejected. In our study the Kruskal Wallis test is used to compare the feature distributions of different walking tasks.

The analysis is performed using the signals pre-processed using the methods described in Section 4.1 but without applying any data augmentation strategy. Furthermore, for the sake of clarity, it is recalled that the free walk, FW, task was performed only in the experiment with older adults. Thus, it is analyzed only for this subjects’ group.

Table 7 and Table 8 show the *p*-values obtained comparing the feature distributions for all the couple of tasks and considering young adults and older adults, respectively. From these results, different observations can be drawn.

First of all, in both the experimental groups considered and in most of the features analyzed, the walking tasks appear significantly different from the baseline B${}_{W}$, with *p*-value usually lower than the significance level. In particular, RMSSD seemed to be the most significant feature in discriminating between walking and resting tasks. On the other hand, this feature presents *p*-values usually greater then α=0.05 when comparing different walking tasks. To this end, it is important to remember that during the baseline acquisition the subjects were still and standing and thus the acquired signals are not affected by the subject movement.

Finally, comparing different walking tasks, features related to peak distance and, in particular, the peak rate, permit to discriminate among most of them. Figure 4 reports the boxplots of the peak rate in case of young adults (left) and older adults (right). From these boxplots, it emerges that subjects’ speed directly affects the heartbeat and, in fact, considering the three different forced speed tasks (F1, F2 and F3), increases in subjects’ speed, caused by consequent increases in peak rate, in particular in the case of young adults. These differences are also noticeable in the *p*-values produced by the Kruskal Wallis test, that are lower than 0.001. Even in the older adults, it is possible to observe a general heartbeat increase related to speed increase (see median values of the F1, F2 and F3 boxes in the right image of Figure 4). This trend is also visible from the *p*-values generated by the Kruskal Wallis test, even if the values produced by the test appear slightly greater than the ones of the young adults. These results may be due to the older adults’ attitude, already described in Section 5.2.1, to walk at a faster pace with respect to the metronome forced speeds.

Moreover, from the analysis of the *p*-values in the case of the older adults, it is not possible to exclude the null hypothesis comparing the free walking task, FW, and the faster of the forced speed tasks, F3. In fact, the *p*-value generated by the test in this comparison with reference to the peak rate appears high (0.67), in accordance with what has already been observed in the EMG analysis.

A comment worth of notice concerns the comparison between the two tasks F3 and WO. For the sake of clarity, we recall that a similar frequency stride has been observed in both the tasks from the EMG analysis. This similarity is also confirmed by the high *p*-values obtained for both young adults and older adults, in most of the PPG features. An interesting exception is observed in the peak rate, while comparing F3 and WO in case of the older adults. In this case, the *p*-value appears lower than α=0.05, and the null hypothesis can be rejected. From the boxplots analysis of the peak rate in case of the older adults (Figure 4 right), it is possible to observe that the median value of the WO box appears higher than the F3 one. This difference may prove that approaching an obstacle generates emotional variations in the older adults, who probably perceive this type of walking as more stressful than the speed-constrained one.

Likewise, it is not possible to reject the null hypothesis between the two tasks of crossing, Obs, and free walking during the collision avoidance task, WO. In fact, in both the populations and in most of the features analyzed, the *p*-values generated by the test are greater than the level of significance considered. The only exception refers again to the peak rate, in case of the older adults, where the *p*-value appeared lower than α = 0.05. In the older adults boxplot of Figure 4 it is also possible to notice that, in general, the heartbeat values observed on task Obs appear higher than the ones of task WO, suggesting that the older adults perceive crossing the zone with the collision avoidance task as a potential stressful task.

#### 5.2.3. Classification of Different Walking Tasks

The aim of this study is to evaluate if different walking tasks, having similar walking pace, could lead to significant changes in the physiological signals and, thus, could be discriminated by a classifier.

For each considered subject group, a different set of analyses is carried out due to slight differences in the experimental protocol, which make it impossible to directly compare all the results achieved in the two populations. Furthermore, only the signals related to heartbeat (PPG) and to sweet gland activity (GSR) are considered in this study. EMG data are excluded from the analysis as we only have data for half of the subjects.

In older adults, three different binary classification tasks are considered, one for each couple of walking activities having similar walking pace (pure free walking FW, free walking during collision avoidance task WO and obstacle crossing Obs). This choice is justified by the results described in Section 5.2.1 concerning the walking cadence analysis. The LOSO Strategy has been used to evaluate the performance of the four classification models: (SVM-Linear, SVM-Gaussian, SVM-Cubic and Cart). The parameters of each classifier have been set as reported in Section 4.2. Similarly to the previous classification analysis, the performances of the trained classifiers are evaluated using accuracy, per-class F1-score and weighted F1-score. For each class, all the signals collected in the dataset are considered in the analysis and, in particular, three different features sets are considered: only PPG features, only GSR features and the join of PPG and GSR features. To this end, we recall that all the PPG and GSR signals collected from one subjects have been discarded as unusable. In addiction, 11 instances related to free walking during collision avoidance task as well as 9 instances related to obstacle crossing task have been removed from the analysis due to their low quality. Please refer to Section 3.2 for more details concerning the classes cardinality.

In Table 9, the results achieved by the different classifiers are summarized, according to the tasks considered, and to the feature set used. From the analysis, it emerges how the physiological signals collected during the FW task seems to be clearly distinguishable both from the WO (free walk part of the collision avoidance task) and the Obs, which is the collision avoidance task itself. In particular, an accuracy of 73% is observed classifying FW from WO, using SVM-Linear and both PPG and GSR. Higher accuracy values are reached in the recognition of FW from Obs. In this case, in fact, the use of PPG features allows to achieve an accuracy of 83% using SVM-Linear. Lower performances, albeit positive, are observed during the recognition of signals collected during the free walking task before or after the obstacle, WO and from signals collected in correspondence to the collision avoiding events, WO. In this case, an accuracy of 68% is achieved with both SVM-Gauss and CART, when considering both PPG and GSR signals. These performances, lower with respect to the others, highlight a greater similarity between the two tasks with reference to the subject’s physiological response and thus his/her stressful state. However, it should be emphasized that this similarity is not so significant to prevent the recognition of the two activities in case of older adults.

The same classification models, evaluation strategy and performance metrics are also adopted in the classification analysis of young adults walking tasks. As already mentioned, the slight difference in the experimental protocol between the two populations makes it not possible to replicate exactly the same analysis. Thereby, in the young adults analysis, we consider only one binary classification task comparing the signals collected during the whole collision avoidance task, WO + Obs (not divided into free walk and obstacle crossing) with the signals acquired during the highest frequency forced speed task (F3). The latter has been selected as representative of a free walking task due to his similarity in terms of frequency stride and because it was selected by the participants as preferred walking frequency among those constrained by the metronome.

The performance achieved by the four classifiers is reported in Table 10. From the analysis of the results, it emerges how the highest accuracy value is reached using the SVM-Linear, when trained with features extracted only from PPG signals. In this case, in fact, an accuracy of 74%, well balanced on both the classes, is observed. The performance achieved using the other two types of feature sets appear, however, positive with *W-F*${}_{1}$ values greater then 65% in almost all the analyses conducted. This proves the ability of the classifiers to well discriminate the two tasks despite the similarity in the participant’s walking cadence.

## 6. Discussion

The positive results achieved from the analysis of data acquired both during the cognitive load and walking sessions demonstrate that physiological signals can be adopted within machine learning models not only to discriminate among different tasks but also to discriminate among different population age.

This allows to answer positively to the research questions Q1 and Q3 and to underline how people of different ages can react differently to similar stimuli, both from an emotional and behavioral point of view.

The performances achieved in the classification of cognitive tasks appear slightly higher than those achieved in the literature in which accuracy values between 60% and 80% are reported [37,86,87]. As a novelty, in our analysis, we sought to apply a previously trained binary classification model to new cognitive tasks. Interesting and promising results are observed in this experiment, demonstrating the ability of the classification model based on physiological data to be generalized on the data collected while performing different tasks. Although these results partially reply to the research question Q2, we need to underline the limits of our analysis. In the present work, we have considered only tasks where arousal is elicited by cognitive load. Future analyses will be performed to include also different types of stressful tasks as well as to consider also activities with movement components (e.g., walking activities).

With reference to the classification performance, PPG seems in general to be more useful in most of the classification tasks; however, the best results are achieved considering both PPG and GSR. Considering in particular the walking session from the results achieved in these experiments it emerges how the physiological signals allow to recognize well the signals collected during different tasks in spite of the walking speed factors. In particular, the analysis have highlighted how the physiological signals of a person tend to change when he/she approaches to a potential dangerous situation as a collision avoidance zone as well as during the crossing of the obstacle itself. Moreover, the good performances of the classification models applied to the signals collected during the walking session demonstrate that there are significant differences in physiological responses with respect to space varying conditions. These considerations answer to the research question Q4.

A potential limit of the dataset collection concerns the subjects that have been selected as participants in the two populations. In fact, all the young adults selected were PhD students or young researchers of the RCAST at the University of Tokyo and, thus, are used to reading difficult texts or performing difficult calculations. On the other hand, the older adults have been selected by a company specialized in experimental subject recruitment and, thus, not representative of a generic elderly population. Of course the low number of instances adopted to train our classifiers can also introduce a bias. However, the cardinality of data collected is similar to other experiments in the state of the art. We have paid particular attention to remove bias related to oversampling as already commented, using a LOSO validation strategy. These results, together with the increasing availability and reliability of wearable devices, are promising in the perspective of the definition of systems that, interacting with subjects, can recognize their emotions and behaviors as well as their age group and consequently adapt. Concerning this topic, several factors such as different cultural aspects or daily habits could be taken into account in future analysis to create systems able to interact with the largest possible number of heterogeneous users. Furthermore, the age of the individuals could also be used as an additional input variable, together with other parameters such as the subject’s health, lifestyle or nutritional habits, in the definition of accurate measures of physiological age that could be used by industrial designers and product developers to guide their work in the development of appropriate technology able to provide efficient and personalized assistance to individuals of different ages and needs.

Another general consideration concerns the classification setting used in the analysis. In the present work, we focused our attention on the analysis of statistical and time domain features, encouraged by the positive results achieved in other works of literature based on PPG and GSR [38,88]. In several works [32,89], the use of these features allowed the authors to reach positive results, with accuracy values around 80%, aligned with our results. In the future, we plan to consider a larger set of features, including frequency [36,90] and time-frequency domain features (such as spectrograms or wavelet coefficients) [91]. Concerning the classifiers considered in our analyses, we adopted SVM and Cart due to their positive performance in the case of moderately unbalanced classes [92,93]. Future analysis will be performed in order to compare the results achieved with these classification models with the ones achievable using different classifiers as ensemble classifiers (e.g., XgBoost or random forest) or neural network. Furthermore, new machine learning techniques will be evaluated such as LSTM models or other deep learning approaches.

Regarding the questionnaires that were filled out by the participants, we started looking at the results obtained by the STAI questionnaires and if they could be effectively correlated to the physiological data of the subjects. The initial results are quite interesting but, at the same time, given how we would need a much larger data cardinality in order to carry out more significant analyses, they cannot be considered significant. For this reason, we intend to concentrate on this kind of data in the future, when we will be able to have a larger amount of data on which perform our tests.

Given the information the data analysis allowed us to gather regarding people’s behavior in response to particular stimuli, especially considering what was obtained from the walking tasks, we started hypothesizing how we could use the results we gathered. One of the research areas we started approaching is the one of agent simulation: the role emotions and affect have in modeling more realistic agent behaviors is gradually being investigated more and more, given the crucial part they play in the interactions a person has with the environment and with other people around him/her. For this reason, we started contemplating how to develop an affective agent model, namely, an affective multiagent system in which the agents’ behavior is also modeled relying on data collected through experiments, both in real-life or controlled laboratory scenarios [94]. Our final goal is to use the information we obtained through the physiological and psychological data analysis to define new variables to include in the agent model, new variables that aim to influence the agents’ behavior in a much more realistic way given how they are based on genuine reactions recorded from real people.

## 7. Conclusions

In their daily life, people are subjected to different stimuli that could affect their behavior and emotions. In particular, the age of a person seems a relevant factor in the definition of how an individual responds to specific stimuli. In this paper, different binary and multi-class classification tasks have proved that physiological signals permit to well discriminate between young adults and older adults, while performing different actions. In particular, the results obtained with the analysis of the EMG during different walking conditions confirm that physiological responses can give significant hints in studying subjects’ behavior and their reactions and confidence within different environments. Moreover, this analysis permits us to underline the different behavior of the older adults with respect to young adults. In particular, the two population groups would need a more specific and in-depth analysis to create ad hoc environments able to meet their needs.

The results reached here can be useful in different analyses; for instance, they may be used to better understand the age-related behavior of subjects in real-world stressful urban environments such as, for example, busy streets or uncomfortable and crowded sidewalks. Moreover, the achieved results suggest that other population groups could be considered, in particular, children or impaired subjects, in order to classify and model their behavior in different real-life conditions, such as while interacting with autonomous vehicles. We were able to demonstrate the generalizability of our model, making physiological signals well suited to be adopted in different conditions. This research paves the way to a new field of research that, for instance, will permit proper forms of communications between pedestrians (especially impaired and frail ones) and autonomous vehicles, taking into account their level of stress. In future scenarios, self-driving vehicles will circulate in urban environments, and will need to adapt to the feelings of pedestrians, being able to provide them effective feedback, properly tuned for the most vulnerable ones.

Simulating a human subject and his/her behavior is an always up-to-date research topic, especially in the artificial intelligence field. Adopting an agent modeling approach often comes as the most natural way to represent a human, since the definition of an agent as a hardware or software system situated in an environment, autonomous, reactive, proactive and social, directly maps to the main features of a human too. The role of emotions and affect in producing more realistic agent behaviors is becoming increasingly crucial, given the important part they play in the interactions a person has with others and with the environment in which he/she lives.Thus in the definition of an affective agent, the influence of age in the emotional perception of the environment should also be modeled.

## Figures and Tables

**Figure 1 sensors-23-03225-f001:**
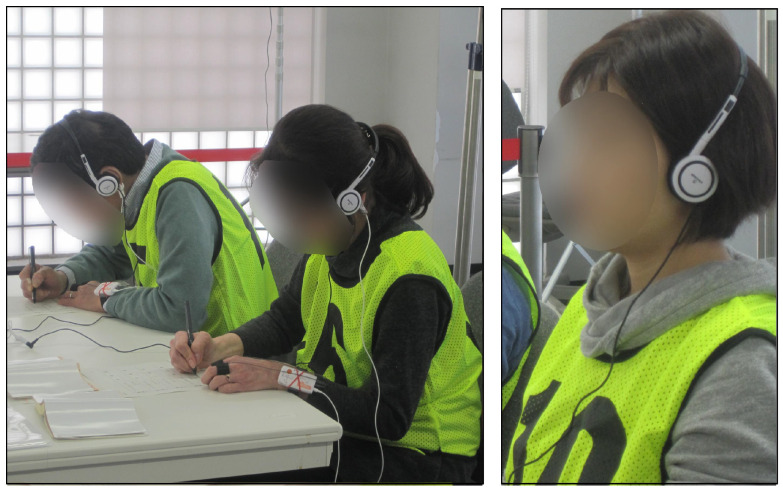
Two subjects during the cognitive load session. **Left**: reporting the results of the math calculation. **Right**: during the audio listening task.

**Figure 2 sensors-23-03225-f002:**
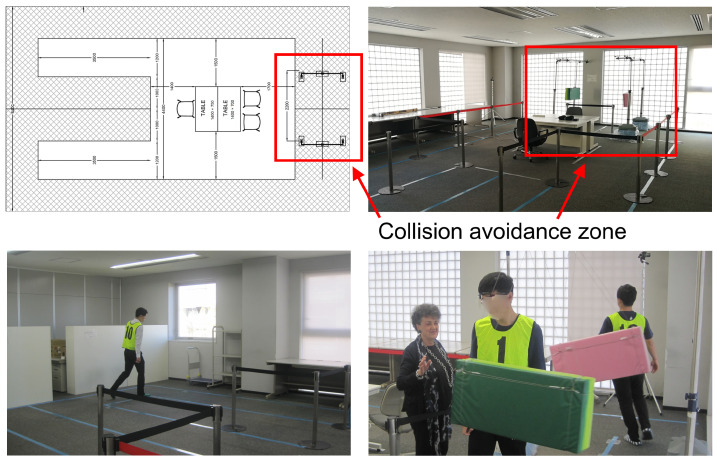
Controlled experiment. Top left: the plant of the laboratory, where the path chosen for the walking activities is depicted. A red rectangle identifies the collision avoidance zone in the two images of the first raw. In this zone, two obstacles are moved by one of the experimenters and the two subjects have to avoid the collision (figure bottom right). During the rest of the path, subjects walk with their own natural pace.

**Figure 3 sensors-23-03225-f003:**
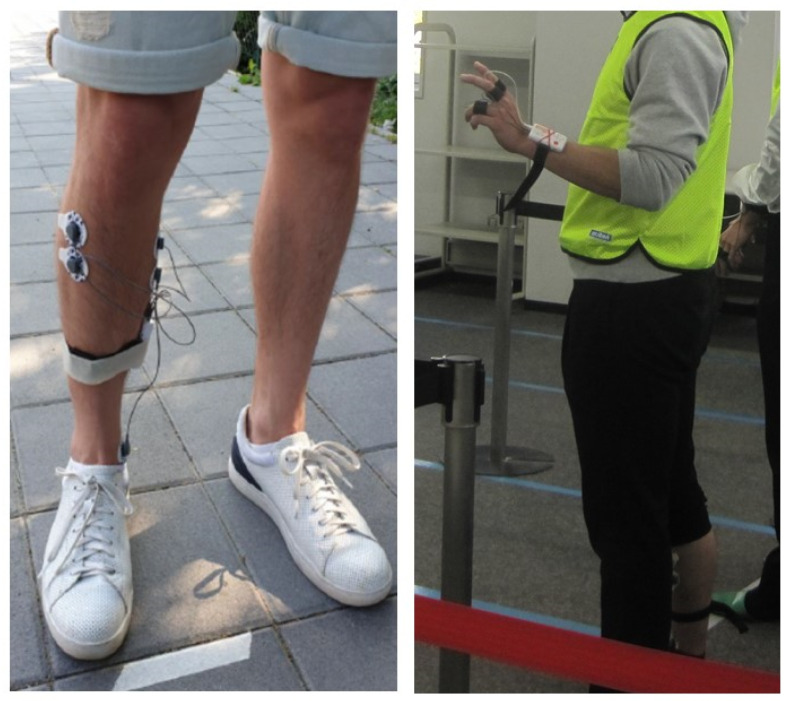
Wearable devices adopted.

**Figure 4 sensors-23-03225-f004:**
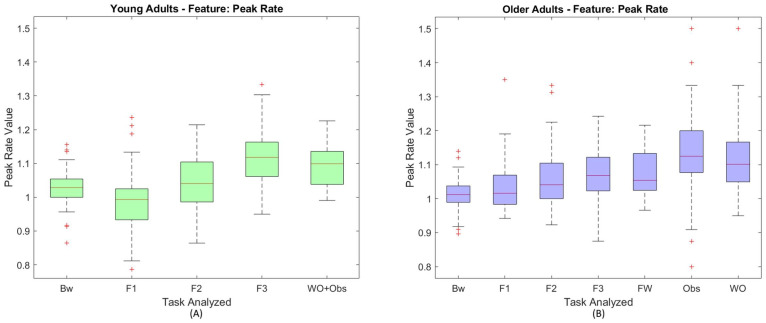
Boxplots of the peak rate in the different tasks. The left boxplot in green (**A**) refers to the young adults while the right one in blue (**B**) refers to the older adults. For each box, the outliers are marked using the ’+’ symbol.

**Table 1 sensors-23-03225-t001:** Number of instances for each task in cognitive load session (first 5 columns) and in walking session (last 8 columns). For each subject group (Young Adults and Older Adults), the first row is related to signals collected using the Shimmer3 GSR+ unit (PPG and GSR) while the second row refers to signals acquired using the Shimmer3 EMG (gastrocnemius muscle EMG and tibial muscle EMG). In the third row is reported the cardinality involved during the classification tasks. In particular, for the reading, comprehension and free walk tasks are reported the cardinality resulting from data augmentation. The analyzed tasks have been referred to according to the following coding: B_*C*_ = baseline task collected during the cognitive load session, R = reading task, C = calculation task, AL = audio listening task, B_*W*_ = baseline task collected during the walking session, F1 = metronome forced speed task (70 bpm), F2 = metronome forced speed task (85 bpm), F3 = metronome forced speed task (100 bpm), FW = pure free walk task, WO = free walk in the collision avoidance task, Obs = obstacle crossing, WO + Obs = single signal for the whole task of collision avoidance (free walk and obstacle crossing).

		Cognitive Load Session					Walking Session							
		B_*C*_	R	C	MC	AL	B_*W*_	F1	F2	F3	FW	WO	Obs	WO + Obs
Young Adults	PPG, GSR	46	32	32	96	96	109	46	46	46	-	-	-	46
	Gastrocn. EMG, Tibial EMG	-	-	-	-	-	55	23	23	23	-	-	-	23
	PPG, GSR (augmented)	46	96	64	96	96	109	46	46	46	-	-	-	46
Older Adlults	PPG, GSR	60	40	40	120	120	129	57	57	57	57	160	104	-
	Gastrocn. EMG, Tibial EMG	-	-	-	-	-	65	28	27	28	28	77	50	-
	PPG, GSR (augmented)	60	120	80	120	120	129	57	57	57	114	160	104	-

**Table 2 sensors-23-03225-t002:** Performance of the binary classifiers in discriminating young adults (Yng) from older adults (Old) in the different tasks analyzed, varying the feature set and adopting a LOSO validation strategy. Three performance metrics are reported: accuracy (Acc.), F1-score ($F_{1}$) and weighted F1-score (W-$F_{1}$). In each table, the W-$F_{1}$ values in bold represent the best performances achieved for each feature set considered, while in red is highlighted the best accuracy at all. The analyzed tasks have been referred according to the following coding: R = reading task, C = comprehension task, MC = math calculation task, AL = audio listening task.

		R Task				C Task				MC Task				AL Task			
	**Classifier**		**Yng**	**Old**			**Yng**	**Old**			**Yng**	**Old**			**Yng**	**Old**	
		**Acc.**	$F_{1}$	$F_{1}$	**W-** $F_{1}$	**Acc.**	$F_{1}$	$F_{1}$	**W-** $F_{1}$	**Acc.**	$F_{1}$	$F_{1}$	**W-** $F_{1}$	**Acc.**	$F_{1}$	$F_{1}$	**W-** $F_{1}$
PPG	SVM Linear	63%	0.53	0.69	62%	**69%**	0.55	0.76	**67%**	**74%**	0.68	0.78	**73%**	60%	0.44	0.69	58%
	SVM Cubic	64%	0.59	0.68	64%	62%	0.60	0.64	62%	70%	0.68	0.72	70%	**64%**	0.62	0.65	**64%**
	SVM Gauss	**66%**	0.61	0.70	**66%**	65%	0.41	0.75	60%	71%	0.65	0.75	71%	56%	0.39	0.66	54%
	Cart	56%	0.54	0.59	57%	59%	0.56	0.62	59%	73%	0.68	0.76	73%	50%	0.46	0.52	50%
	Random	50%	0.47	0.52	50%	50%	0.44	0.55	50%	50%	0.47	0.52	50%	50%	0.47	0.52	50%
GSR	SVM Linear	**63%**	0.54	0.70	**63%**	**66%**	0.63	0.69	**66%**	64%	0.59	0.68	64%	**67%**	0.60	0.72	**67%**
	SVM Cubic	58%	0.53	0.62	58%	58%	0.53	0.61	58%	**66%**	0.61	0.70	**66%**	58%	0.51	0.63	58%
	SVM Gauss	62%	0.54	0.68	62%	65%	0.59	0.70	65%	62%	0.55	0.67	62%	59%	0.49	0.66	58%
	Cart	59%	0.52	0.64	59%	63%	0.57	0.67	62%	54%	0.49	0.59	54%	57%	0.54	0.61	58%
	Random	50%	0.47	0.52	50%	50%	0.44	0.55	50%	50%	0.47	0.52	50%	50%	0.47	0.52	50%
PPG and GSR	SVM Linear	**75%**	0.71	0.77	**74%**	**69%**	0.67	0.72	**70%**	**78%**	0.75	0.81	**78%**	**65%**	0.59	0.69	**65%**
	SVM Cubic	64%	0.62	0.66	64%	61%	0.56	0.65	61%	69%	0.64	0.72	68%	64%	0.58	0.69	64%
	SVM Gauss	72%	0.67	0.67	72%	61%	0.56	0.65	61%	71%	0.68	0.74	71%	**65%**	0.59	0.69	**65%**
	Cart	65%	0.60	0.69	65%	58%	0.52	0.63	58%	67%	0.63	0.70	67%	62%	0.55	0.66	61%
	Random	50%	0.47	0.52	50%	50%	0.44	0.55	50%	50%	0.47	0.52	50%	50%	0.47	0.52	50%

**Table 3 sensors-23-03225-t003:** Confusion matrix of SVM-Linear for the multi-class recognition task. Six classes are considered, one for each couple of the cognitive load task: math calculation MC, reading R and audio listening AL, and subject age: young adult (Yng), and older adults (Old). The values in bold are the main diagonal elements and represent the cases where the classes predicted by the classifier and true classes agree.

		Predicted Class					
		**MC_Yng**	**R_Yng**	**AL_Yng**	**MC_Old**	**R_Old**	**AL_Old**
**True class**	MC_Yng	**56%**	7%	3%	19%	10%	4%
	R_Yng	4%	**46%**	2%	5%	26%	17%
	AL_Yng	0%	6%	**50%**	0%	4%	40%
	MC_Old	14%	7%	1%	**71%**	1%	7%
	R_Old	2%	15%	2%	3%	**76%**	3%
	AL_Old	2%	12%	18%	2%	2%	**66%**

**Table 4 sensors-23-03225-t004:** Performance of the binary classifiers in discriminating the instances collected during the math calculation tasks MC from the instances collected during the audio listening task AL, varying the feature set and adopting a LOSO validation strategy. Three performance metrics are reported: accuracy (Acc.), F1-score ($F_{1}$) and weighted F1-score (W-$F_{1}$). In each population group, the W-$F_{1}$ values in bold represent the best performances achieved for each feature set considered, while in red is highlighted the best accuracy at all.

		PPG Features				GSR Features				PPG and GSR Features			
	**Classifier**		**MC**	**AL**			**MC**	**AL**			**MC**	**AL**	
		**Acc.**	* $F_{1}$ *	$F_{1}$	* **W-** * $F_{1}$	**Acc.**	$F_{1}$	$F_{1}$	**W-** $F_{1}$	**Acc.**	$F_{1}$	$F_{1}$	**W-** $F_{1}$
Young Adults	SVM Linear	**79%**	0.77	0.80	**79%**	**93%**	0.93	0.94	**93%**	**91%**	0.91	0.91	**91%**
	SVM Cubic	72%	0.72	0.72	72%	91%	0.91	0.92	91%	80%	0.80	0.79	80%
	SVM Gauss	76%	0.75	0.76	76%	91%	0.90	0.91	91%	83%	0.84	0.83	83%
	Cart	64%	0.63	0.64	64%	**93%**	0.93	0.93	**93%**	**91%**	0.91	0.91	**91%**
	Random	50%	0.50	0.50	50%	50%	0.50	0.50	50%	50%	0.50	0.50	50%
Older Adults	SVM Linear	**80%**	0.80	0.81	**80%**	**92%**	0.92	0.92	**92%**	**90%**	0.90	0.90	**90%**
	SVM Cubic	72%	0.71	0.73	72%	90%	0.90	0.91	90%	89%	0.89	0.89	89%
	SVM Gauss	78%	0.78	0.78	78%	**92%**	0.92	0.92	**92%**	89%	0.89	0.89	89%
	Cart	75%	0.75	0.74	75%	88%	0.88	0.89	88%	**90%**	0.90	0.90	**90%**
	Random	50%	0.50	0.50	50%	50%	0.50	0.50	50%	50%	0.50	0.50	50%

**Table 5 sensors-23-03225-t005:** Percentage of instances for each cognitive load task classified as high arousal and low arousal using pre-trained SVM-Linear classifiers. Each classifier has been trained considering the instances collected during the MC task as high arousal and the instances collected from AL as low arousal. The classifiers are then applied to classify the following: baseline task B${}_{C}$, reading task R and comprehension task C. In the first three rows, the results generated using the signals collected from young adults (Yng) are reported, while the last three rows are related to signals acquired from the older adults participants (Old). In each population group and cognitive load task, the values in bold represent the highest percentage of instances between high and low arousal for each feature set considered.

		PPG		GSR		PPG + GSR	
		**% Instances** **High Arousal**	**% Instances** **Low Arousal**	**% Instances** **High Arousal**	**% Instances** **Low Arousal**	**% Instances** **High Arousal**	**% Instances** **Low Arousal**
Yng	B${}_{C}$	17%	**83%**	17%	**83%**	17%	**83%**
	R	34%	**66%**	28%	**72%**	22%	**78%**
	C	**69%**	31%	**84%**	16%	**75%**	25%
Old	B${}_{C}$	7%	**93%**	10%	**90%**	8%	**93%**
	R	50%	50%	35%	**65%**	40%	**60%**
	C	**53%**	47%	**83%**	17%	**73%**	27%

**Table 6 sensors-23-03225-t006:** The frequency strides estimated for both subject groups are reported and compared with the metronome frequencies (F1, F2 and F3).

	Young Adults	Older Adults
**Metronome**	**EMG**	**EMG**
F1 = 0.58	0.59	0.66
F2 = 0.70	0.72	0.76
F3 = 0.83	0.85	0.85

**Table 7 sensors-23-03225-t007:** Kruskal Wallis *p*-values: comparison between tasks in young adults. The values highlighted in red refer to *p*-value lower than the significance level chosen: α=0.05. The analyzed tasks are: B${}_{W}$ = baseline task acquired in walking session, F1 = metronome forced speed task (70 bpm), F2 = metronome forced speed task (85 bpm), F3 = metronome forced speed task (100 bpm), WO + Obs = single signal for the whole task of collision avoidance (free walk and obstacle crossing).

First Task	Second Task	Maximum	Minimum	Mean	Variance	Peak Rate	IBI	RMSSD
B${}_{W}$	F1	<0.001	<0.001	0.29	<0.001	<0.001	<0.001	<0.001
B${}_{W}$	F2	<0.001	<0.001	0.21	<0.001	0.24	0.45	<0.001
B${}_{W}$	F3	<0.001	<0.001	0.03	<0.001	<0.001	<0.001	<0.001
B${}_{W}$	WO + Obs	<0.001	<0.001	0.01	<0.001	<0.001	<0.001	<0.001
F1	F2	0.58	0.66	0.80	0.64	<0.001	<0.001	0.14
F1	F3	0.77	0.64	0.28	0.14	<0.001	<0.001	0.70
F1	WO + Obs	<0.001	<0.001	0.18	<0.001	<0.001	<0.001	<0.001
F2	F3	0.51	0.91	0.42	0.37	<0.001	<0.001	0.08
F2	WO + Obs	<0.001	<0.001	0.20	0.01	<0.001	<0.001	0.08
F3	WO + Obs	<0.001	<0.001	0.71	0.17	0.10	0.24	<0.001

**Table 8 sensors-23-03225-t008:** Kruskal Wallis *p*-values: comparison between tasks in older adults. The values highlighted in red refer to *p*-values lower than the significance level α=0.05. The analyzed tasks are: B${}_{W}$ = baseline collected during walking session F1 = metronome forced speed task (70 bpm), F2 = metronome forced speed task (85 bpm), F3 = metronome forced speed task (100 bpm), FW = pure free walk task, WO = free walk in the collision avoidance task, Obs = obstacle crossing.

First Task	Second Task	Maximum	Minimum	Mean	Variance	Peak Rate	IBI	RMSSD
B${}_{W}$	F1	<0.001	<0.001	<0.001	<0.001	0.18	0.15	<0.001
B${}_{W}$	F2	<0.001	<0.001	<0.001	<0.001	<0.001	<0.001	<0.001
B${}_{W}$	F3	<0.001	<0.001	<0.001	<0.001	<0.001	<0.001	<0.001
B${}_{W}$	FW	<0.001	<0.001	0.20	<0.001	<0.001	<0.001	<0.001
B${}_{W}$	Obs	0.19	<0.001	0.01	<0.001	<0.001	<0.001	<0.001
B${}_{W}$	WO	<0.001	<0.001	<0.001	<0.001	<0.001	<0.001	<0.001
F1	F2	0.78	0.89	0.98	0.66	0.03	0.03	0.56
F1	F3	0.90	0.60	0.73	0.49	<0.001	<0.001	0.30
F1	FW	0.28	<0.001	0.01	0.78	<0.001	<0.001	0.13
F1	Obs	0.07	0.49	0.88	0.10	<0.001	<0.001	0.13
F1	WO	0.38	0.74	0.01	0.08	<0.001	<0.001	0.31
F2	F3	0.90	0.64	0.82	0.73	0.16	0.18	0.58
F2	FW	0.38	0.01	0.01	0.82	0.38	0.27	0.30
F2	Obs	0.04	0.32	0.99	0.19	<0.001	0.02	0.41
F2	WO	0.30	0.85	0.02	0.15	<0.001	0.02	0.77
F3	FW	0.25	0.02	<0.001	0.60	0.67	0.84	0.73
F3	Obs	0.07	0.19	0.92	0.35	<0.001	0.23	0.91
F3	WO	0.36	0.82	0.04	0.33	0.03	0.36	0.66
FW	Obs	0.01	<0.001	0.05	0.14	<0.001	0.14	0.77
FW	WO	0.06	<0.001	<0.001	0.09	0.01	0.20	0.35
Obs	WO	0.18	0.18	0.05	0.94	0.01	0.72	0.51

**Table 9 sensors-23-03225-t009:** Performance of the binary classifiers in discriminating walking tasks having similar walking pace considering the signals collected from older adults. Three tasks are compared: pure free walk FW, free walk before and after the collision avoidance zone WO and obstacle crossing Obs. The analysis are performed varying the feature set used (PPG, GSR or PPG and GSR) and adopting a LOSO validation strategy. Three performance metrics are reported: accuracy (Acc.), F1-score ($F_{1}$) and weighted F1-score (W-$F_{1}$). The W-$F_{1}$ values in bold represent the best performances achieved for each feature set considered, while in red is highlighted the best accuracy at all.

		WO vs. FW				FW vs. Obs				WO vs. Obs			
	**Classifier**		**WO**	**FW**			**Obs**	**FW**			**Obs**	**WO**	
		**Acc.**	$F_{1}$	$F_{1}$	**W-** $F_{1}$	**Acc.**	$F_{1}$	$F_{1}$	**W-** $F_{1}$	**Acc.**	$F_{1}$	$F_{1}$	**W-** $F_{1}$
PPG	SVM Linear	68%	0.73	0.61	68%	**83%**	0.80	0.85	**83%**	**66%**	0.45	0.75	**63%**
	SVM Cubic	**70%**	0.75	0.62	**70%**	70%	0.67	0.72	70%	55%	0.37	0.65	54%
	SVM Gauss	67%	0.74	0.54	66%	76%	0.71	0.80	76%	64%	0.32	0.75	58%
	Cart	64%	0.71	0.52	63%	71%	0.69	0.72	71%	55%	0.40	0.64	54%
	Random	50%	0.54	0.45	50%	50%	0.51	0.49	50%	50%	0.55	0.44	51%
GSR	SVM Linear	**64%**	0.70	0.53	**63%**	77%	0.74	0.79	77%	65%	0.30	0.76	58%
	SVM Cubic	58%	0.62	0.54	59%	80%	0.79	0.82	80%	64%	0.50	0.72	63%
	SVM Gauss	62%	0.69	0.51	62%	**81%**	0.79	0.82	**81%**	**66%**	0.44	0.76	**63%**
	Cart	63%	0.69	0.56	63%	74%	0.73	0.74	74%	58%	0.47	0.65	58%
	Random	50%	0.54	0.45	50%	50%	0.51	0.49	50%	50%	0.55	0.44	51%
PPG and GSR	SVM Linear	71%	0.75	0.66	71%	81%	0.78	0.83	80%	67%	0.46	0.77	65%
	SVM Cubic	66%	0.71	0.59	66%	80%	0.78	0.82	80%	64%	0.51	0.71	63%
	SVM Gauss	**73%**	0.77	0.65	**72%**	**81%**	0.80	0.83	**81%**	68%	0.52	0.76	66%
	Cart	64%	0.71	0.53	63%	69%	0.67	0.71	69%	**68%**	0.56	0.74	**67%**
	Random	50%	0.54	0.45	50%	50%	0.51	0.49	50%	50%	0.55	0.44	51%

**Table 10 sensors-23-03225-t010:** Performance of the binary classifiers in discriminating tasks having similar walking pace considering the signals collected from young adults. Two tasks have been compared: the metronome forced speed task of 100 bpm F3 and the collision avoidance task WO+Obs, including, in a single instance, both the free walking before and after the collision avoidance zone and the obstacle crossing. The analysis are performed varying the feature set used (PPG, GSR or PPG and GSR) and adopting a LOSO validation strategy. Three performance metrics are reported: accuracy (Acc.), F1-score ($F_{1}$) and weighted F1-score (W-$F_{1}$). The W-$F_{1}$ values in bold represent the best performances achieved for each feature set considered, while in red is highlighted the best accuracy at all.

	PPG Features				GSR Features				PPG and GSR Features			
**Classifier**		**F3**	**WO + Obs**			**F3**	**WO + Obs**			**F3**	**WO + Obs**	
	**Acc.**	$F_{1}$	$F_{1}$	**W-** $F_{1}$	**Acc.**	$F_{1}$	$F_{1}$	**W-** $F_{1}$	**Acc.**	$F_{1}$	$F_{1}$	**W-** $F_{1}$
SVM Linear	**74%**	0.76	0.71	**74%**	65%	0.61	0.69	65%	**72%**	0.72	0.71	**72%**
SVM Cubic	60%	0.62	0.57	60%	65%	0.67	0.64	65%	62%	0.65	0.59	62%
SVM Gauss	72%	0.75	0.68	71%	**66%**	0.61	0.70	**66%**	68%	0.69	0.68	68%
Cart	63%	0.67	0.58	62%	63%	0.67	0.59	63%	62%	0.65	0.58	62%
Random	50%	0.50	0.50	50%	50%	0.50	0.50	50%	50%	0.50	0.50	50%

## Data Availability

The dataset generated with the experiments together with the MATLAB codes to extract the features of the physiological signals here described and analyzed can be found at: https://drive.google.com/drive/folders/1pQ2z7kEHEteN3HLtdd2XzffFxsdkWdd4?usp=sharing (accessed on 1 March 2023).

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
