# Peer review of "Behavior and Task Classification Using Wearable Sensor Data: A Study across Different Ages†"

_sensors, 2023, doi:10.3390/s23063225_

Round 1

Reviewer 1 Report

The authors focus on task classification starting from physiological signals acquired using wearable sensors. Classification between different ages and the performed task is demonstrated. The paper is well organized. And sufficient experimental data is offered and analyzed. Nevertheless, the authors should consider the following suggestions to improve the paper.

1.     In the manuscript, volunteers with ages from 24 to 65 are chosen. Is it possible to use the model to discriminate children and adults?

2.     Will the health condition influence the results of experiment?

3.     There are some mistakes in formatting (e.g., in line 292).

Author Response

We thank Reviewer 1 for taking the time to provide us with comments and suggestions to improve our manuscript. In the following is reported a point-by-point response to the Reviewer’s comments.

Comments and Suggestions for Authors

The authors focus on task classification starting from physiological signals acquired using wearable sensors. Classification between different ages and the performed task is demonstrated. The paper is well organized. And sufficient experimental data is offered and analyzed. Nevertheless, the authors should consider the following suggestions to improve the paper.

  1. In the manuscript, volunteers with ages from 24 to 65 are chosen. Is it possible to use the model to discriminate children and adults?

According to the obtained results we believe that our models could be able to discriminate also between children and adults. However, as we had not yet performed an experiment with children we do not have data to test them. We plan in the future to also consider this population. 

We have added the following comment in the conclusions: 

“Moreover the achieved results suggest that other population groups could be considered, in particular children or impaired subjects, in order to classify and model their behavior in different real life conditions, such as while interacting with autonomous vehicles”. 

  • Will the health condition influence the results of experiment?

In general, subjects’ health conditions could influence the results of the analysis. For this reason, during the experiment, we have checked, using a proper questionnaire, that all the participants were healthy and have no mental or heart diseases. And this was an inclusion criterion. We report the following phrase in the manuscript:

 “Mental or heart diseases could influence the physiological responses, thus the healthy conditions were considered as inclusion criteria for the selection of the participants.” 

  1. There are some mistakes in formatting (e.g., in line 292).

We have checked and corrected typos. Thank you for pointing it out. 

Reviewer 2 Report

The reviewer has some minor questions need to be answered in the manuscript text

  1. What motivated the choice of physiological signals for classification in this study?
  2. How were the cognitive load tasks designed, and how did they differ from the walking conditions?
  3. What kind of normalization was performed on the physiological signals to account for subject variability?
  4. Can you provide more information on the feature extraction process used in this study?
  5. What validation strategies were employed to evaluate the classification models?
  6. How was the generalizability of the binary classification models tested?
  7. What were the key challenges encountered in denoising the raw physiological data?
  8. Were there any significant differences in the physiological responses of young and older adults during the experiments?
  9. How were the different walking conditions designed, and what kind of obstacles were included in the environment?
  10. Were there any unexpected findings or results that emerged during the data analysis process?
  11. Can you discuss any limitations or potential sources of bias in the study design or data collection process?
  12. How does this study build upon or contribute to existing research on physiological signal analysis and affective state recognition?
  13. What implications do the results of this study have for the design of systems that can automatically react to human behaviors and emotions?
  14. What are the main strengths of the CLAWDAS dataset, and how might it be useful for future research in this area?
  15. What are some potential avenues for future research based on the findings of this study?

Author Response

We thank the reviewer for his/her valuable comments. 

We uploaded a file with the detailed answers to all of them. 

Reviewer 3 Report

Attached Separately

Author Response

We thank Reviewer 3 for taking the time to read our paper carefully and we really appreciated his/her nice comments about the value of our research.